# Placental lesions in stillbirths: A case-control study using the Amsterdam criteria and predictive models at a UK tertiary unit

**Brenda F. Narice**[1,2]*, **Victoria Byrne**[1], **Joanna Shepherd**[2], **Marta C. Cohen**[1], **Dilly O. Anumba**[1,2]

**1** Division of Clinical Medicine, School of Medicine and Population Health, University of Sheffield, Sheffield, United Kingdom, **2** Sheffield Teaching Hospitals NHS Foundation Trust, Sheffield, United Kingdom

* b.f.narice@sheffield.ac.uk

## Abstract

### Introduction

The UK stillbirth rate remains higher than in many high-income countries, with placental disorders -particularly maternal vascular malperfusion (MVM) lesions -linked to adverse maternal and fetal outcomes. This study examines placental lesions in stillbirth at one of the largest maternity units in the UK using the Amsterdam criteria for histological classification. It also retrospectively examines whether women with global/partial MVM – where most maternal decidual vessels show pathological changes but are only partially occluded- would have received aspirin and further surveillance if their placental dysfunction risk had been assessed using the Fetal Medicine Foundation (FMF) algorithm from the Tommy's app in their first trimester.

### Materials and methods

We conducted a case-control study of spontaneous non-anomalous stillbirths (≥24 weeks) at Sheffield maternity unit from 2018 to 2021 (n = 83). We then compared singleton stillbirths at term with matched livebirths. Placental lesions were categorised with the Amsterdam criteria. Using the FMF's algorithm which has only been recently introduced in our unit, we then retrospectively calculated the risk for placental dysfunction in women who experienced preterm PET stillbirth and also in those whose placentas showed global/partial MVM.

### Results

MVM was the most common placental lesion in stillbirths, significantly more frequent than in livebirths (p < 0.001). The FMF algorithm had higher predictive accuracy for PET than the traditional NICE model in stillbirths [AUC: 0.76 (95% CI 0.65–0.86) vs 0.51 (95% CI 0.39–0.63), p = 0.03], **but only when at least one continuous**

**Data availability statement:** All relevant data are within the manuscript and its Supporting Information files.

**Funding:** BFN salary is partly covered by a National Institute for Health and Care Research [NIHR] Academic Clinical Lectureship [O&G CL-2020-04-003]. No other funding was received and the funders had no role in the study.

**Competing interests:** None.

**Abbreviations:** AUC, area under the curve; CIUE, chronic intervillositis of unknown etiology; IUGR, intrauterine growth restriction; FIR, fetal inflammatory response; FMF, Fetal Medicine Foundation; FVM, fetal vascular malperfusion; MIR, maternal inflammatory response; MVM, maternal vascular malperfusion; OR, odds ratio; PET, preeclampsia; VUE, villitis of unknown etiology.

**variable such as PAPP-A was included**. In women with stillbirth and whose placentas showed global/partial MVM, first-trimester placental risk assessment using the FMF algorithm during the first trimester would have identified most of them as high risk [FMF AUC: 0.7 (0.58–0.80), p = 0.02].

## Conclusion

MVM is frequently found in stillbirths. Our retrospective placental dysfunction risk assessment suggests that Tommy's algorithm would have more accurately identified women who went onto experience stillbirth with significant MVM lesions as high risk, leading to aspirin treatment and closer monitoring. Further research is needed to confirm these findings and potentially enhance placental dysfunction screening to reduce stillbirth rates.

---

## Introduction

The stillbirth rate in England and Wales, approximately 4.1 per 1,000 total births annually, has remained relatively constant over the past decades and is higher than in other high-income countries. Recognising the urgency of this issue, the UK Department of Health & Social Care set a goal in 2020 to halve the stillbirth rate by 2025 [1]. Not yet in track, achieving this ambitious target needs a better understanding of the underlying causes of stillbirths, particularly placental issues, which account for over 30% of cases according to the recent *Mothers and Babies: Reducing Risk through Audit and Confidential Enquiries* (MBRRACE-UK) data [2].

To improve consistency in diagnosing placental lesions, the Amsterdam Consensus was established in 2016, providing standardised histopathological terminology and diagnostic criteria. The Amsterdam criteria identify distinct patterns of placental injury, including maternal vascular malperfusion (MVM), fetal vascular malperfusion (FVM), acute chorioamnionitis which might comprise a maternal inflammatory (MIR) and fetal inflammatory response (FIR), villitis of unknown etiology (VUE) and delayed villous maturation (DVM) [3,4]. This standardisation is crucial for reliable data comparison and effective intervention strategies.

MVM is the most common placental lesion found in stillbirths. It is linked to an inadequate blood supply to the feto-placental unit and is frequently observed in preeclampsia (PET), a leading cause of stillbirth. MVM is also more common in pregnancies affected by intrauterine growth restriction (IUGR), which continues to be prevalent in stillbirth cases even when congenital anomalies and infections are excluded. [5]. Understanding the relationship between MVM and adverse perinatal outcomes is crucial due to the risk of recurrence in future pregnancies [6].

Various antenatal models have been proposed to predict phenotypic presentations of placental dysfunction, mostly PET, such as the National Institute for Health and Care Excellence -NICE- criteria and the Fetal Medicine Foundation -FMF-, [7,8]. The NICE criteria rely on a binary risk assessment of major and moderate risk factors. The FMF model, on the contrary, integrates continuous variables with biophysical

and biochemical markers to generate a weighted risk score for placental dysfunction. Both models aim to identify high-risk women who may benefit from prophylactic interventions such as aspirin, supported by the ASPRE trial findings which showed it might reduce preterm PET risk and potentially other placental disorders [9,10].

In Sheffield, the FMF model -integrated into the Tommy's app- has replaced the NICE criteria for assessing the risk of placental dysfunction since 2022. However, it remains unclear whether the FMF model predicts only PET, or if it can also identify other conditions associated with placental dysfunction, which may share common risk factors. Global/partial MVM features, for example, are linked to a 25% recurrence rate, a crucial factor for informing current and future pregnancy management [6,11].

This study aims to determine the frequency of placental lesions associated with stillbirths using the Amsterdam criteria in one of the largest maternity units in the UK and evaluate the effectiveness of FMF models in predicting PET and MVM with clinical significance. These findings could improve targeting of intervention and in turn, reduce stillbirth rates.

## Materials and methods

### Case and control selection

We conducted a descriptive analysis of all registered spontaneous stillbirths (>24 weeks) at Sheffield Jessop Wing Maternity Centre between January 2018 and December 2021. Cases involving termination of pregnancy and known congenital anomalies were excluded. The study period was chosen based on the accessibility of electronically stored patient records, which have been locally available since the end of 2017. Data was accessed between March 30th and June 30th 2023 as per the NHS registration form. Stillbirth rates at Sheffield during this period were slightly higher than the national average (~5 per 1,000 births) but remained stable compared with previous years with no significant difference in our local trust during the COVID-19 pandemic period.

For singleton stillbirths at term (≥ 37 weeks), we further conducted a gestational age and ethnicity matched case-control study. There were not enough placentas available at other gestational ages to create a control group with conditions independent of the Amsterdam criteria. The control group at term was selected from the same maternity unit between August and December 2021, based on the accessibility of placental reports from relatively low-risk livebirths. During the COVID-19 pandemic, all women who tested positive for SARS-CoV-2 had their placentas sent for pathology, regardless of obstetric outcome. SARS-CoV-2 infection in unvaccinated women has been associated with an increased stillbirth risk [12]. However, placentas and placental function markers from vaccinated women who experienced asymptomatic COVID-19 are unlikely to have been affected by the infection [13]. Therefore, for the control group, we selected placentas from women with no comorbidities, fully vaccinated with at least two Pfizer doses 14 days before testing positive, and asymptomatic at the time of COVID-19 screening, which was routinely performed in the UK at each face-to-face healthcare interaction. None of the included placentas displayed features suggestive of COVID-19 infection, such as perivillous fibrin deposition, chronic histiocytic intervillositis, and trophoblast necrosis which would have warranted further assessment by immunohistochemistry and/or in situ hybridization. [12,14].

### Ethics and confidentiality

The research was approved as a service evaluation by the Sheffield Teaching Hospital (STH) Clinical Effectiveness Unit (AIMS 11530/2023), following the NHS Health Research Authority Toolkit Criteria. All women information was anonymised and encrypted to ensure confidentiality.

### Data collection and handling of missing data

Maternal socio-economic demographics and relevant clinical history were collected and coded for analysis. Postcodes were converted into an Index of Multiple Deprivation (IMD) score using the IMD Postcode Checker [15]. Placental reports

were completed by perinatal pathologists at Sheffield Children's Hospital, following the Amsterdam consensus. Other lesions associated with stillbirth, but not included in the Amsterdam criteria, were also considered. The cause of stillbirth was assessed using the ReCoDe classification system [4,16]. For multiple pregnancies, stillbirth was defined as the loss of any fetus, and all such cases were included in the analysis. Postmortem information was unavailable as a different hospital Trust manages it and it is not electronically linked to the mother's records. Efforts were made to retrieve missing information by contacting the regions where women were originally booked.

## Placental analysis

Placental examination was conducted in two sequential assessments: macroscopic and microscopic. The macroscopic assessment included placental weight (used to determine the fetomaternal weight ratio), placental disk dimensions, umbilical cord description (diameter, length, site of insertion, and coiling index) and membrane evaluation. Fetoplacental weight ratio and centiles were calculated based on gestational age following the references in *Placental Pathology* [17]. The umbilical cord index was further calculated using de Laat, Franx [18] guidance.

Microscopically, placental lesions were assessed using the Amsterdam criteria. In addition, some relevant entities associated with stillbirth but not included in the consensus were also assessed to ensure comprehensive coverage of all histopathological findings and to highlight any potential limitations of the Amsterdam criteria. FVM lesions were further classified into high and low grade according to Redline and Ravishankar [19], and MVM into global/ complete (early: distal villous hypoplasia/ late: accelerated villous) and segmental/partial (villous infarcts) for the predictive models [6].

## Statistical analysis

Descriptive and inferential statistics were performed in SPSS 29 (IBM, US). For normally distributed variables, parametric tests were used and presented as mean and standard deviation (SD). For non-normally distributed variables, non-parametric tests were applied, and data presented as median and interquartile range. A Chi-square test was used for discrete variables (linear by linear association), with results presented as raw numbers and percentages (%). To measure association between stillbirth/livebirth and placental lesions, odds ratio ± 95% CI were calculated.

## Placental dysfunction antenatal risk assessment

First-trimester risk assessment for PET and placental dysfunction was retrospectively evaluated using both the NICE criteria and the FMF algorithm across all stillbirth and control cases. Although these criteria were originally designed to assess gestational hypertensive disorders, the FMF algorithm is now a primary tool for evaluating placental dysfunction as employed in the Tommy's app [11].

The NICE criteria consist of major and moderate risk factors. A single major risk factor, such as Type 1 or Type 2 diabetes, chronic hypertension, renal disease, autoimmune disorders, or a personal history of pre-eclampsia (PET), places a woman at high risk. Moderate risk factors include maternal age ≥ 40 years, BMI ≥ 35 kg/m², nulliparity, a pregnancy interval of ≥ 10 years, and multiple pregnancies. The presence of two moderate risk factors also classifies a woman as high risk [7].

The first trimester FMF algorithm uses multiple regression to assess PET risk. Criteria include pregnancy characteristics (type and dating), maternal characteristics (demographic, medical, and clinical variables), biophysical measurements (arterial pressure and uterine artery pulsatility index [PI]), and biochemical measurements (placental growth factor and pregnancy-associated plasma protein A [PAPP-A]) [20]. Unlike the NICE criteria, which use discrete risk categories, the FMF criteria apply continuous variables (e.g., age and weight) in risk assessment.

Although the FMF algorithm can be applied using only clinical information, albeit with an increased false positive rate, in this study, we only included cases with at least one further biophysical and/or biochemical variable available.

The predictive performance was assessed with sensitivity, specificity, positive predictive value (PPV), negative predictive value (NPV), and Area under the Receiver Operating Characteristic Curve (AUC) on MedCalc 22.009 (MedCalc Software, Ostend, Belgium).

## Results

### Stillbirth cohort

Data was obtained from the Jessop Wing maternity unit in Sheffield for the cohort of women who had experienced a stillbirth (≥24 weeks) between 2018–2021 (n = 130). Cases of terminations of pregnancy (TOPs) and congenital anomalies were excluded (n = 40). From the remaining 90 eligible cases, seven were removed as no placental reports were available, (Fig 1). The retrospective cohort consisted of 83 cases, including 72 (86.7%) singletons, 8 (9.6%) twins and 3 (3.6%) triplets.

The average maternal age at booking was 30.5 years (y) ± 6.75 y, with significantly higher maternal age for twin pregnancies (34.4 y ± 7.6 y) and triplet pregnancies (30.5 y ± 6.7 y) compared to singleton pregnancies (29.6 y ± 6.2 y), $p < 0.002$.

Most women who experienced stillbirth were multiparous, non-smokers, without a history of substance misuse and had conceived spontaneously (Table 1). Over half of all women who had a stillbirth resided in the lower postcode deciles [1–3], which correspond to the most deprived areas, with a significant association ($p < 0.05$). A further Chi-square test revealed a correlation between ethnicity and postcode deciles ($p < 0.02$), showing that most women who self-identified as BAMER (Black, Asian, Minority Ethnic, and Refugee) lived in the poorer areas of Sheffield. Women residing in the more deprived areas were also more likely to be late bookers ($p < 0.02$).

Two thirds of the stillbirths were premature and almost half of all cases were small for gestational age (<10th centile), (Table 1).

Over a third of the stillbirth for which gross pathology was available (n = 79) had a placenta weight under the 10th centile (n = 30, 38%). These were significantly associated with a fetoplacental weight ratio above 90th (n = 11, 13.9%, $p < 0.001$).

The commonest histopathological placental lesion in the stillbirth cohort was MVM either in isolation (n = 36, 42%) and/ or combined (n = 44, 53%) followed by high grade FVM (n = 11, 13%) and inflammatory lesions mostly stage 3 necrotizing MIR/FIR lesions (n = 5, 6%) (Fig 2). There was as well a case of SARS-CoV-2 placentitis registered among the stillbirth

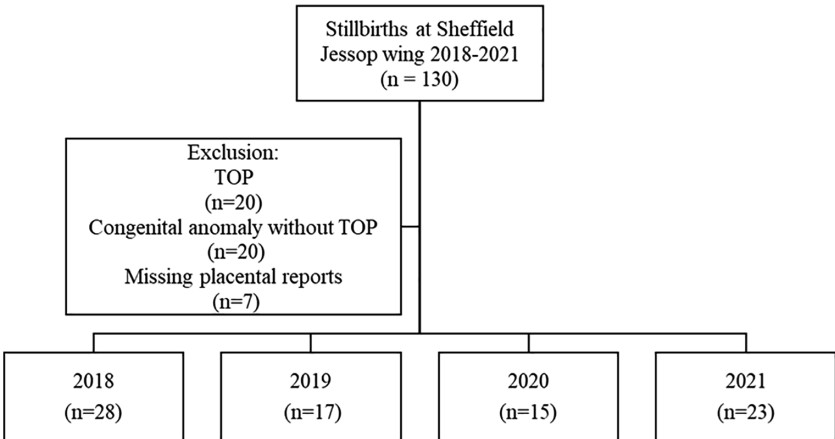

**Fig 1. Cohort of spontaneous stillbirth pregnancies at Jessop Wing between 2018-2021 included in quantitative analysis (n = 83).** *TOP: termination of pregnancy.*

**Table 1. Clinical demographic characteristics of the stillbirth cohort.**

| Maternal Characteristics | Stillbirth |
|---|---|
| Age, years | 30.50 (±0.6.7) (n = 82) |
| BMI kg/m$^2$ | 25.08 (22.0–30.9) (n = 72) |
| Gravidae | 3 (1.0–5.0) (n = 80) |
| Parity | 1 (0.0–2.7) (n = 80) |
| Ethnicity (n, %) <br> - White <br> - Black <br> - Asian <br> - Mixed <br> - Other | (n = 77) <br> 43 (55.8%) <br> 11 (14.3%) <br> 13 (16.9%) <br> 3 (3.9%) <br> 7 (9.1%) |
| Cigarette smoker (n,%) | 11(15.1%) (n = 73) |
| Substance abuse (n, %) | 7 (9.9%) (n = 71) |
| Deprivation Index (n, %) <br> - 1–3 (most deprived areas) <br> - 4–6 <br> - 7–10 (least deprived areas) | (n = 82) <br> 45 (54.9%) <br> 21 (25.6%) <br> 16 (19.5%) |
| Conception Type (n, %) <br> - Spontaneous <br> - ART | (n = 76) <br> 73 (96.1%) <br> 3 (3.9%) |
| Gestation Type (n, %) <br> - Singleton <br> - Twin <br> - Triplet | (n = 83) <br> 72 (86.7%) <br> 8 (9.6%) <br> 3 (3.6%) |
| Previous stillbirth (n,%) | 3 (3.4%) (n = 75) |
| Gestational age at booking (weeks)* | 8.9 (7.4–11.57) (n = 60) |
| Gestational age at delivery (weeks) <br> - Preterm (<37 weeks) <br> - Term | 33 (28.58–37.86) (n = 83) <br> 55 (66.3%) <br> 28 (33.7%) |
| Fetal weight centile adjusted by GA using Inter-growth calculator | 10.1 (0.7–45.20) (n = 83) |
| Fetal weight centile < 10th centile | 41 (49.4%) (n = 83) |

For continuous variables, data is expressed as mean ± SD or median and interquartile range (25th-75th). Discrete variables are presented as raw number and percentage. ART: assisted reproductive technology, BMI: body mass index., GA: gestational age, NVD: normal vaginal delivery.

(n = 1, 1.2%). Within the combined MVM group with detailed histologic information (n = 40), almost half displayed villous infarcts (47.5%) followed by accelerated villous maturation (40%) and decidual arteriopathy (12.5%).

### Control-matched singleton stillbirth at term

Singleton stillbirths at term were further compared to gestational age- and ethnicity-matched controls. Maternal age at booking was slightly higher among women who had a stillbirth, but this difference did not reach statistical significance (30.1 ± 5.4 years vs 29.3 ± 4.0 years, p = 0.4). No differences were seen for other potential confounding factors such as smoking and substance misuse. However, pregnant women who had a stillbirth were more likely to be multiparous, reside in the more deprived regions of South Yorkshire and had a small for gestational age fetus (defined as an estimated weight of <10th centile) than their livebirth counterparts (p = 0.01), (Supplementary, S1 Table).

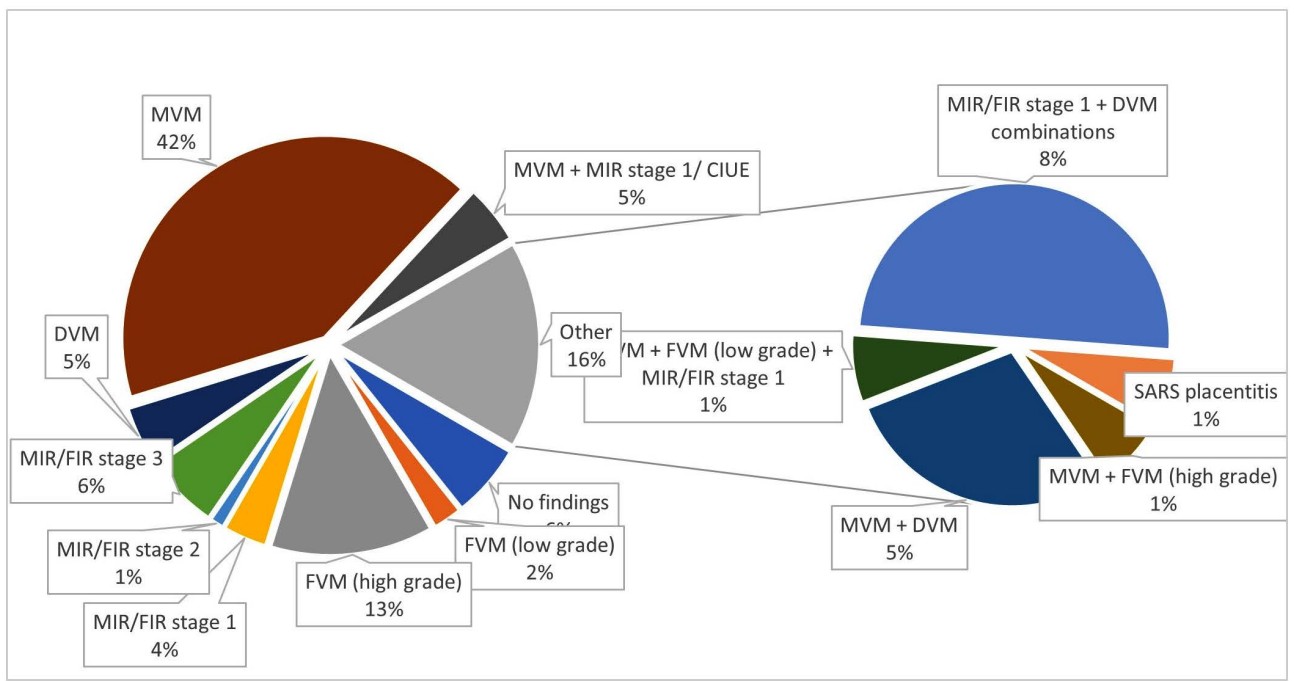

**Fig 2. Percentage of placental lesions observed in the spontaneous stillbirth (>24 weeks gestation) cohort at Sheffield Jessop Maternity Wing between 2018-2021 (n = 83).** *CIUE: chronic intervillositis of unknown etiology, DVM: delayed villous maturation, Fetal inflammatory response, FMV: fetal vascular malperfusion, MVM: maternal vascular malperfusion, MIR: maternal inflammatory response.*

Stillbirth placentas were noted to be smaller (<10th centile, p < 0.01) with higher fetoplacental weight ratio than the live-birth (>90th centile, p = 0.03), (Table 2).

At term, MVM remained the most prevalent placental lesion reported in stillbirths either in isolation or combined with other placental lesions (46.4%), (Table 2). It was also significantly more common in stillbirths than in livebirths (p < 0.001), (Supplementary, S1 and S2 Figs). High grade FVM were also significantly higher in stillbirths whereas the opposite was true for low grade FVM (p = 0.02 and <0.001 respectively), (Table 1, Supplementary S2(b) Fig). Placentas with no relevant findings and with MIR stage 1, on the other hand, were significantly more frequent in livebirths (p = 0.04 and p < 0.001 respectively). No differences were seen in VUE and DVM (Table 2, Supplementary S2(c) and S2(d) Fig).

## Cause of death using the ReCoDe classification system

Based on medical records and death certificates, using the ReCoDe classification system for stillbirths, conditions from group C (placenta) accounted for over one-third of cases (31.3%, n = 26). The most common condition was placental abruption, observed in approximately 20% of stillbirths, followed by intrauterine growth restriction and placental insufficiency.

A quarter of stillbirths remained unexplained at birth (26.5%, n = 22). Among these, 26 placental lesions were identified, either in isolation or combination. The most common lesions in the unexplained group at birth were MVM (34.6%, n = 9), high-grade FVM (26.9%, n = 7), MIR/FIR stage 1 and 2 (26.9%, n = 7), low-grade FVM (7.7%, n = 2), and CIUE (3.8%, n = 1).

## Antenatal prediction of PET and gestational hypertension disorders

Of all stillbirths with a placental report (n = 83), 18.1% (n = 15) were from pregnancies complicated by either PET or severe gestational hypertension. Remarkably, no cases of PET were identified without concurrent MVM.

**Table 2. Placental lesions observed in isolation or in combination with other lesions, in stillbirth and live-birth cases (n = number of placentas examined).**

| Placental gross pathology and histopathology | Stillbirth (n = 28) % (n=) | Livebirth (n = 28) % (n=) | p-value |
|---|---|---|---|
| Placental weight <10th centile | 39.3% (n = 11) | 10.7% (n = 3) | **0.01** |
| Fetoplacental weight ratio >90th centile | 21.4% (n = 6) | 3.6% (n = 1) | **0.03** |
| Umbilical cord coiling index<br>- Hypocoiling (<0.07 coil/cm)*<br>- Hypercoiling (>0.3 coil/cm)* | 3.8% (n = 1)<br>19.2% (n = 5) | 21.4% (n = 6)<br>0.0% | 0.8<br>0.3 |
| MVM | 13 46.4% (n = 13) | 3.6% (n = 1) | **<0.001** |
| FVM high grade | 17.9 (n = 5) | 0.0% | **0.02** |
| FVM low grade | 3.6% (n = 1) | 14.3% (n = 4) | **<0.001** |
| MIR<br>- Stage 1<br>- Stage 2<br>- Stage 3 | 25% (n = 7)<br>3.6% (n = 1)<br>14.3% (n = 4)<br>7.1% (n = 2) | 11 39.3% (n = 11)<br>7 25% (n = 7)<br>10.7% (n = 3)<br>3.6% (n = 1) | N/S<br>**0.02**<br>N/S<br>N/S |
| FIR<br>- Stage 1<br>- Stage 2<br>- Stage 3 | 14.3% (n = 4)<br>3.6% (n = 1)<br>7.1% (n = 2)<br>3.6% n = 1 | 21.4% (n = 6)<br>10.7% (n = 3)<br>7.1% (n = 2)<br>3.6% (n = 1) | N/S<br>N/S<br>N/S<br>N/S |
| VUE high grade | 3.6% (n = 1) | 7.1% (n = 2) | N/S |
| VUE low grade | 0.0% | 10.7% (n = 3) | N/S |
| DVM | 21.4% (n = 6) | 21.4% (n = 6) | N/S |
| No significant findings | 0.0% | 32.1% (n = 9) | **<0.001** |

*For continuous variables, data is expressed as mean ±SD or median and interquartile range (25th-75th). Discrete variables are presented as raw number and percentage. N/S: not significant, DVM: delayed villous maturation, FMV: fetal vascular malperfusion, FIR: fetal inflammatory response, MIR: maternal inflammatory response, MVM: maternal vascular malperfusion, VUE: villitis of unknown etiology. \*Available for n = 26 only.*

We first retrospectively compared the performance of the NICE criteria against the FMF algorithm to retrospectively predict PET and gestational hypertension in cases and controls. Due to incomplete biophysical and/or biochemical data, this comparative analysis was only possible for half of the stillbirth cases (55.4%, n = 46) and almost all controls (89.3%, n = 25). The FMF algorithm demonstrated higher sensitivity in predicting PET compared to the NICE criteria [75.0% (95% CI 34.9–96.8) vs 12.5% (0.3–52.7), respectively], though at the cost of reduced specificity [64.06% (51.1–75.7) vs 89.1% (78.8–95.5), respectively] (Supplementary, S2 Table). Overall, the FMF algorithm showed superior predictive performance, with an AUC of 0.76 (95% CI 0.65–0.86) compared to the NICE criteria's AUC of 0.51 (0.39–0.63), p = 0.03 (Fig 3).

### Antenatal prediction of MVM and MVM specific phenotypes

Using the same dataset, we assessed the FMF algorithm's predictive performance for all MVM lesions (n = 24) as well as for those with specific features: global/complete (n = 11) and segmental/partial (n = 13). Although no statistical significance was found for antenatal prediction of all MVM combined (p = 0.07), the algorithm significantly predicted the occurrence of MVM with global/total features which are associated with 25% recurrence [FMF AUC: 0.7 (0.58–0.80), p = 0.02], (Fig 4, Supplementary S3 Table).

### Discussion

Our study identified MVM as the most prevalent placental lesion in stillbirth cases in Sheffield, occurring significantly more often in stillbirths than in livebirths (p < 0.001). High-grade FVM was also more frequently observed in stillbirths compared

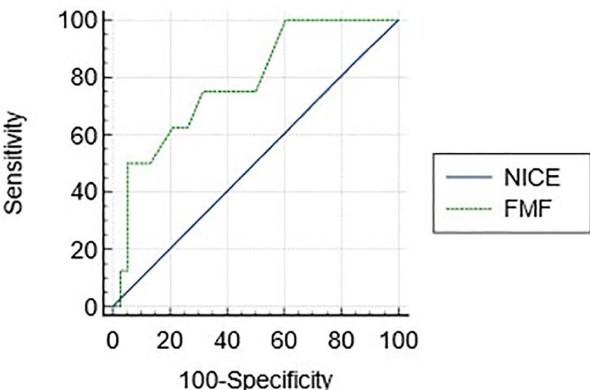

**Fig 3. Receiver operating characteristics (ROC) curves for the prediction of PET and gestational hypertensive disorders comparing NICE criteria and the FMF algorithm (p = 0.03).** *FMF: Fetal Medicine Foundation, NICE: National Institute for Health and Care Excellence.*

to livebirths, though it was less common than MVM. These were also the most prevalent placental lesions observed in the stillbirths whose cause of demise could not be determined at death. These findings are critical as placental histopathology offers valuable insights into the underlying causes of fetal death helping to reduce the number of unexplained stillbirth cases [21].

Our findings also emphasise the importance of considering the fetoplacental weight ratio and its centile during placental analysis. In our cohort, a fetoplacental weight ratio above the 90th percentile, i.e., cases of disproportionally small placentas to normal fetal weights, was statistically significant between stillbirths and livebirths at term. In these instances, placental insufficiency due to MVM and reduced placental volume likely reached a critical point, leading to fetal death. This highlights the need for tools that can accurately assess placental volume and function to identify pregnancies with significantly suboptimal placentas but no evidence of intrauterine growth restriction.

MIR stage 1 lesions were significantly more common in livebirths than stillbirths, which may be linked to the labour process rather than infections [22]. Inflammatory lesions are considered to be a continuum, with more advanced stages and severity of maternal inflammation associated with higher perinatal mortality [23]. Of note, no significant differences were observed for higher MIR stages or any FIR stages between index cases and controls.

We used the FMF algorithm to retrospectively assess its performance in predicting PET and MVM lesions. Despite limitations due to missing data, the FMF algorithm significantly outperformed the NICE criteria in predicting PET, with an AUC of 0.76 (p = 0.002) in alignment with extensive literature supporting the externally validated algorithm [24]. Interestingly, in our study, the FMF algorithm was also capable of predicting specific MVM features, in particular the global/partial phenotype, which is associated with high risk of recurrence and might be amenable to interventions including pre-conceptual health optimisation as well as potential antenatal therapies such as aspirin and closer surveillance in pregnancy [6]. However, as MVM can also occur independently of pre-eclampsia in other forms of placental dysfunction, these findings should be interpreted with caution, since MVM is not specific to hypertensive disorders of pregnancy. This predictive capability could potentially guide interventions in current and future pregnancies [25].

Differences in predictive performance between the FMF multimodal model and the NICE criteria suggest that antenatal care for many pregnancies ending in stillbirth prior to 2022 might have been managed differently if the FMF algorithm had been applied, although this may not have altered outcomes. Importantly, the advantage of the FMF model was observed only when at least one continuous variable (blood pressure or PAPP-A) was included, In the absence of continuous variables, FMF and NICE produced relatively similar results for history-based PET screening. The FMF algorithm was fully implemented in Sheffield only in 2022 as part of the Tommy's app initiative [26].

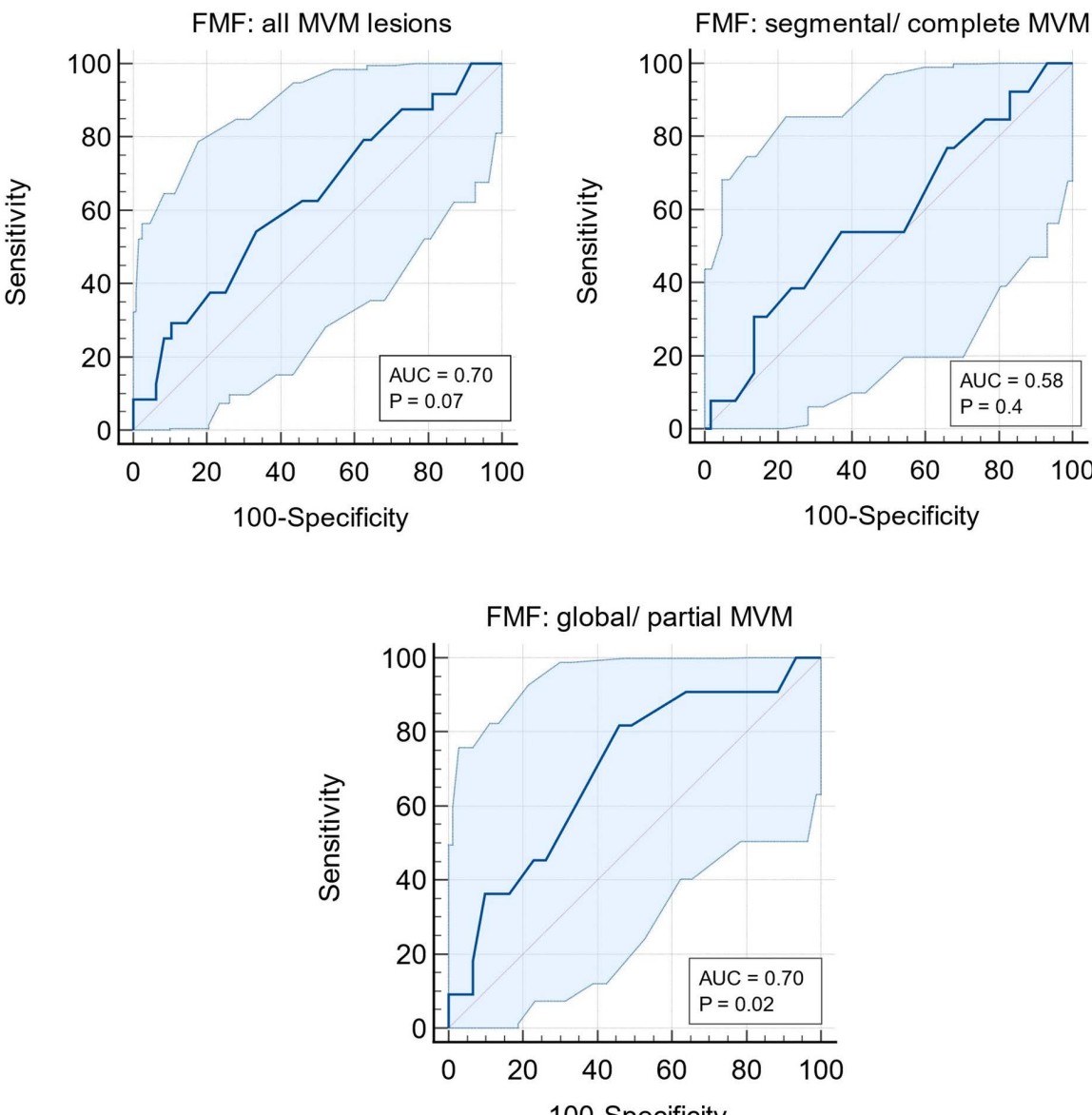

**Fig 4. Receiver operating characteristics (ROC) curves for the prediction of (a) all MVM, (b) segmental/complete MVM and (c) partial/global MVM lesions.** *FMF: Fetal Medicine Foundation, MVM: maternal vascular malperfusion, p<0.05 significant.*

A major strength of our cohort study is its comprehensiveness as it encompasses the entire population of spontaneous stillbirths with accessible placental reports in Sheffield making the sample representative. All placentas were examined at a single laboratory following a standardised protocol. Additionally, Jessop Wing was an early adopter of Tommy's app which relies on the FMF algorithm [26].

A notable limitation of this retrospective study is the challenge of control selection. Identifying an appropriate control group of livebirths was difficult as placental histopathology reports from uncomplicated pregnancies were needed. However, placentas from uneventful pregnancies are not routinely referred for examination, with priority given to cases involving maternal comorbidities and perinatal complications [27,28].

For the term group, we constructed a gestational age- and ethnicity-matched control using placental reports from fully vaccinated, asymptomatic women who had livebirths during the same period as the stillbirth cases and whose only indication for placental analysis was a positive SARS-CoV-2 test during pregnancy. While this might raise concerns, studies have shown that double vaccination with the Pfizer vaccine mitigates adverse outcomes such as stillbirths, and SARS-CoV-2 does not affect the placenta [12,29,30]. We selected women who received their vaccination at least two weeks before birth, and whose placentas showed no signs of SARS-CoV-2 pathology. Despite adjusting for numerous confounding variables, we were unable to match the control group for deprivation index scores. This may be multifactorial as women who experience stillbirth in the UK are more likely to live in deprived areas as highlighted by MBRRACE reports. Additionally, women agreeing to vaccination tend to be healthier and reside in less deprived areas [31]. Although maternal age was slightly higher among term stillbirths, this difference was not statistically significant, likely reflecting the limited sample size given the well-established impact of advanced maternal age on stillbirth risk [32].

Stillbirths can occur at any gestational age, and we recognise that gestational age-matched controls for preterm stillbirths would have provided valuable insights. However, placentas from preterm livebirths are only referred for histopathological examination in our unit if linked to specific clinical conditions such as chorioamnionitis, abruption, etc. These conditions are typically associated with specific Amsterdam criteria and may have biased the results. It would also have been informative to examine placentas from liveborn SGA infants to determine whether similar MVM patterns are present in this group, particularly as approximately 40% of our term stillbirths had fetal weights below the 10th centile. Although such analyses were not possible due to the limited availability of placentas from livebirths, previous studies have shown that MVM lesions are also common in placentas from SGA pregnancies, even in the absence of hypertension, supporting the concept that MVM reflects a shared pathway of placental dysfunction [33,34]. It is also worth noting that, although the matched control group was retrieved only from August-December 2021 rather than the entire period of index cases, no significant differences were observed in stillbirth and livebirth rates during those years and hence, it should still be representative of the whole period.

Not all studies specified whether the placental lesions were causative or contributory. Additionally, in the absence of postmortem information, it was not possible to determine that placental lesions were the sole -rather than contributory- cause of stillbirth in every case. Furthermore, twin and triplet pregnancies carry an increased risk of stillbirth, which may be influenced by chorionicity [35]. Given the small number of multiple pregnancies in our cohort, separate analysis was not feasible, but we have acknowledged this as a potential source of bias.

Antenatal care practices have evolved in recent years. Uterine Doppler measurements in the first trimester were not introduced at Jessop Wing until 2022, and PAPP-A was only measured when women consented to first-trimester trisomy screening. As a result, the performance of the FMF algorithm, which we retrospectively recreated, may have been suboptimal in some cases due to partial missing biophysical and biochemical data, an issue now resolved with routine testing in Sheffield. Similarly, the performance of the NICE criteria may have been underestimated, as all women deemed high-risk by the algorithm were offered aspirin, likely improving placental development and reducing hypertensive disorders. These limitations are inherent to the retrospective study design and may have introduced residual bias that could not be fully accounted for, although they reflect contemporaneous clinical practice during the study period.

## Conclusion

Understanding the etiology of stillbirths is essential for optimising antenatal care and reducing stillbirth rates. Our study identifies MVM as the most common placental lesion linked to stillbirth, suggesting that improved antenatal risk stratification and targeted interventions may be beneficial. Furthermore, our findings emphasise the need for techniques to monitor placental volume and function to improve screening and intervention.

## Supporting information

**S1 Table. Case-control characteristics for singleton term pregnancies.**
(ODT)

**S1 Fig. The frequency of placental lesions either in isolation or combined observed in singleton stillbirths (n = 28) and livebirths (n = 28) delivered at term.** *Statistically significant (p < 0.05), FIR: fetal inflammatory response, DVM: delayed villous maturation, FMV: fetal vascular malperfusion, MVM: maternal vascular malperfusion, MIR: maternal inflammatory response, VUE: villitis of unknown etiology.
(ODT)

**S2 Fig. Forest plot demonstrating the odds ratio of (a) MVM, (b) FVM, (c) MIR and FIR, (d) VUE and (e) DVM in stillbirth and livebirth placentas.** DVM: delayed villous maturation, FMV: fetal vascular malperfusion, FIR: fetal inflammatory response, MIR: maternal inflammatory response, MVM: maternal vascular malperfusion, VUE: villitis of unknown etiology.
(ODT)

**S2 Table. Clinical performance of PET and gestational hypertension risk assessment in the stillbirth cohort.**
(ODT)

**S3 Table. Clinical performance of MVM risk assessment in the stillbirth cohort using the FMF algorithm embedded in the Tommy's. app.**
(ODT)

**S1 Data. Raw data for analysis.** This file contains the raw data used for the statistical analyses presented in the manuscript. All data have been anonymized.
(XLSX)

## Acknowledgments

We would like to thank the Sheffield Teaching Hospitals CEU for critically reviewing the protocol, Ms Mariam Labib for supporting data collection, and all the women who chose Jessop Wing for their pregnancies.

## Author contributions

**Conceptualization:** Brenda F. Narice, Marta C. Cohen.

**Data curation:** Brenda F. Narice, Victoria Byrne, Joanna Shepherd, Marta C. Cohen.

**Formal analysis:** Brenda F. Narice, Victoria Byrne, Marta C. Cohen.

**Investigation:** Brenda F. Narice.

**Methodology:** Brenda F. Narice, Victoria Byrne.

**Project administration:** Brenda F. Narice.

**Resources:** Dilly O. Anumba.

**Supervision:** Brenda F. Narice, Marta C. Cohen, Dilly O. Anumba.

**Validation:** Brenda F. Narice.

**Writing – original draft:** Brenda F. Narice.

**Writing – review & editing:** Brenda F. Narice, Victoria Byrne, Marta C. Cohen, Dilly O. Anumba.

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
