## [Decision Letter · Decision Letter 0]

24 Sep 2025

Dear Dr.  Narice,

Thank you for submitting your manuscript to PLOS ONE. After careful consideration, we feel that it has merit but does not fully meet PLOS ONE’s publication criteria as it currently stands. Therefore, we invite you to submit a revised version of the manuscript that addresses the points raised during the review process.

We look forward to receiving your revised manuscript.

Kind regards,

Nishel Mohan Shah, PhD

Academic Editor

PLOS ONE

Journal Requirements:

“BFN salary is partly covered by a National Institute for Health and Care Research Academic Clinical Lectureship. No other funding from the public, commercial or not-for-profit sectors was received for this work.”

4. Please note that funding information should not appear in any section or other areas of your manuscript. We will only publish funding information present in the Funding Statement section of the online submission form. Please remove any funding-related text from the manuscript.

5. We note that your Data Availability Statement is currently as follows:

“All relevant data are within the manuscript and its Supporting Information files.”

6. Please amend your manuscript to include your abstract after the title page.

Reviewer's Responses to Questions

**Comments to the Author**

1. Is the manuscript technically sound, and do the data support the conclusions?

Reviewer #1: Yes

Reviewer #2: Yes

2. Has the statistical analysis been performed appropriately and rigorously?

Reviewer #1: Yes

Reviewer #2: Yes

3. Have the authors made all data underlying the findings in their manuscript fully available?

Reviewer #1: Yes

Reviewer #2: Yes

4. Is the manuscript presented in an intelligible fashion and written in standard English?

Reviewer #1: Yes

Reviewer #2: Yes

Reviewer #1: Thank you for inviting me to review this manuscript.

This retrospective case-control study investigates histopathological placental findings in non-anomalous stillbirth compared vs livebirths in a UK tertiary centre from 2018 to 2021.

The topic is relevant, especially considering that UK stillbirth rates remains among the highest of high-income countries, despite a good and continued efforts to reduce perinatal mortality as per The Saving Babies’ Lives Care Bundle’ (SBLCB) version 3. Please, do cite SVBCBv3.

Considering the UK stillbirth rate in 2024 was 4.1 per 1,000 births, a sample size of 83 stillbirths is relatively large, thus basic statistical analysis for binary and continuous variables is scientifically sound. Particularly, the association of stillbirths with social deprivation, histological maternal vascular malperfusion and high fetoplacental ratio when compared to livebirths has been effectively stressed, also in regard to MBRRACE-UK.

Although some major considerations need to be added.

- Can authors please specify the rate of stillbirth/births per year in Sheffield? This is to compare with current national rates (current stillbirth rate in UK is about 4,1 per 1000 births), especially because the sample size has been extracted from COVID pandemic period where stillbirth rates reportedly increased. Since the authors have discussed COVID extensively, including vaccinated controls only is a justified approach to mitigate potential bias.

- For preterm stillbirths, matching placentas by gestational age was not feasible due to limited sample availability of livebirths, although two-thirds of the stillbirths were preterm. However, for term births, matching by maternal age should have been attempted and this limitation should be acknowledged in the discussion as a potential source of bias, in view of the huge impact of advanced maternal age on stillbirth risk.

- I appreciate the effort to validate the FMF and NICE algorithms for predicting PET in a retrospective cohort of stillbirths versus matched controls. While MVM is present in all PET cases, MVM can also occur independently of PET, meaning that MVM can exist in non-hypertensive pregnancy disorders/placental dysfunction.

It would have been informative to examine placentas in livebirths of SGAs to explore whether MVM is present in these cases as well, particularly because approximately half of your stillbirths fetal weight is <10th centile. Although I appreciate it may not have been possible due to paucity of placentas, I believe authors should comment on literature findings of IUGR/SGAs placentas in regard to their findings.

- Twin and triplet pregnancies carry an increased risk of stillbirth per se, which may be further affected by chorionicity. Ideally, these cases should be analyzed separately; however, I understand that this may have been challenging. Please cite this as a potential source of bias and clarify whether "stillbirth" refers to the loss of one or both twins in the dataset.

Minor:

- Please state clearly it is a retrospective case-control study, both in abstract and in manuscript.

- The FMF algorithm outperformed NICE in predicting PET only when at least one continuous variable (e.g., MAP, UtA-PI, or PAPP-A) was included. In the absence of continuous variables, FMF and NICE are basically the same in terms of purely history-indicated PET screening. Please clarify both in abstract and manuscript.

-line 16, 'placental disorders' should be changed to placental histopathological lesions

Reviewer #2: This is a well-conducted retrospective cohort study that provides valuable insights into the role of placental pathology, particularly maternal vascular malperfusion (MVM) in stillbirths. The use of standardized Amsterdam criteria and the comparison between NICE and FMF risk assessment models are strengths. The study highlights the potential of early risk stratification to improve antenatal care and reduce stillbirth rates.

This study includes all eligible spontaneous stillbirths from a large tertiary center over a four-year period, making the findings representative of that specific population. The findings have direct implications for antenatal care, suggesting that improved risk assessment could lead to timely interventions (e.g., aspirin prophylaxis). The study accounted for confounding factors such as ethnicity, deprivation index, and COVID-19 vaccination status and the authors are also to be commended for their transparent and thorough discussion of the study's limitations, particularly concerning the control group and retrospective data collection.

The manuscript is clearly written, logically structured (Introduction–Methods–Results–Discussion–Conclusion), and well illustrated with figures and tables. It meets the standards of PLOS ONE.

Limitations and Areas for Improvement

1.Inherent Limitations of Retrospective Design: The study is constrained by data completeness. As the authors note, the retrospective "reconstruction" of the FMF risk score was necessary due to missing first-trimester biophysical and biochemical markers in many cases, which may have impacted the accuracy of its calculated performance. Furthermore, clinical interventions during the study period (e.g., aspirin administration based on NICE criteria) could introduce a bias that is difficult to account for retrospectively

2.Challenges in Control Group Selection: The authors rightly identify control selection as a major challenge. A perfect "low-risk" control group for stillbirths is difficult to obtain, as placentas from uncomplicated pregnancies are not routinely sent for pathology.

3.Limited Sample Size: The total sample size of 83 stillbirths is relatively small for subgroup analyses, such as predicting PET (n=15) or specific MVM types. This can limit statistical power and leads to wide confidence intervals in some analyses, reducing the certainty of the findings.

**Do you want your identity to be public for this peer review?** For information about this choice, including consent withdrawal, please see our Privacy Policy

Reviewer #1: **Yes: ** Silvia L Spinillo

Reviewer #2: No

---

## [Author Response · Author response to Decision Letter 1]

15 Oct 2025

Sheffield , UK

October 9th, 2025

Dear Dr Nishel Mohan Shah, PhD

Editor PLOS ONE Journal

Ref: PONE-D-25-32249. Placental lesions in stillbirths: a retrospective study using the Amsterdam criteria and predictive models at a UK tertiary unit.

Many thanks for reviewing our manuscript and for accepting it for potential publication subject to revisions. We would also like to express our appreciation to Reviewers 1 (R1) and 2 (R2) for their time and thoughtful suggestions, which helped improve the manuscript.

Below, we outline the changes we have made in response to your comments and those of the reviewers.

Editor, point #1:

Thank you for the reminder. We have reviewed and updated the manuscript and all associated files to ensure they meet PLOS ONE’s style and file-naming requirements.

Editor, point #2:

We note that the grant information you provided in the ‘Funding Information’ and ‘Financial Disclosure’ sections do not match.

Thank you for bringing this to our attention. We have reviewed and corrected the grant information to ensure consistency between the Funding Information and Financial Disclosure sections. The correct grant numbers for the awards supporting BFN’s time for the study (NIHR CL-2020-04-003) are now listed accurately in the online Funding Information section.

Editor, point #3:

Thank you for stating the following financial disclosure:

“BFN salary is partly covered by a National Institute for Health and Care Research Academic Clinical Lectureship. No other funding from the public, commercial or not-for-profit sectors was received for this work.”

Thank you for your comment. We confirm that the funders had no role in study design, data collection and analysis, decision to publish or preparation of the manuscript. This statement has been included in the online form.

Editor, point #4:

Please note that funding information should not appear in any section or other areas of your manuscript. We will only publish funding information present in the Funding Statement section of the online submission form. Please remove any funding-related text from the manuscript.

Thank you for the clarification. We have removed all funding-related text from the manuscript. Funding information is now included only in the Funding Statement section of the online submission form, in accordance with PLOS guidelines.

Editor, point #5:

We note that your Data Availability Statement is currently as follows:

“All relevant data are within the manuscript and its Supporting Information files.”

Yes, we can confirm that our submission contains all raw data required to replicate the results of the study. The raw data are provided alongside each reported mean and standard deviation in the manuscript, together with the statistical tests employed for analysis. All relevant data, including the minimal data set as defined by PLOS (comprising the data required to replicate all study findings, as well as related metadata and methods), are contained within the manuscript and its Supporting Information files (appendix).

Editor, point #6:

Please amend your manuscript to include your abstract after the title page.

Thank you for noting this. We have amended the manuscript to include the abstract immediately following the title page, in accordance with PLOS ONE formatting requirements.

Editor, point #7:

We have reviewed the relevant literature and added citations where appropriate to support statements in the manuscript. These added references are detailed later in this rebuttal letter and we have ensured that all citations included are directly relevant to the points discussed.

Reviewer 1

We thank Reviewer 1 for their careful reading of our manuscript and for the constructive comments provided. We appreciate the detailed feedback on both major and minor points and have addressed all suggested amendments in our revised manuscript.

Reviewer 1, major point #1:

Can authors please specify the rate of stillbirth/births per year in Sheffield? This is to compare with current national rates (current stillbirth rate in UK is about 4,1 per 1000 births), especially because the sample size has been extracted from COVID pandemic period where stillbirth rates reportedly increased. Since the authors have discussed COVID extensively, including vaccinated controls only is a justified approach to mitigate potential bias.

Thank you for raising this point. Local data from dashboards show that Sheffield Teaching Hospitals had a stillbirth rate of about 5 per 1,000 births during the study period, which was slightly higher than the national average. The rate was generally stable compared with pre-COVID years. After the CQC visit in 2021, several improvements were introduced, including a care bundle and tools such as Tommy’s app in 2022–2023, which helped reduce stillbirth rates in later years (now around 3.7 per 1,000). Following your advice, we have clarified this point in the manuscript and added information about stillbirth trends before and during the study [Lines 125-128].

Reviewer 1, major point #2:

For preterm stillbirths, matching placentas by gestational age was not feasible due to limited sample availability of livebirths, although two-thirds of the stillbirths were preterm. However, for term births, matching by maternal age should have been attempted and this limitation should be acknowledged in the discussion as a potential source of bias, in view of the huge impact of advanced maternal age on stillbirth risk.

Thank you for your insightful comment regarding maternal age in stillbirth matched for term births. We have now examined maternal age at booking for term stillbirths and livebirths. While maternal age was higher among the stillbirths, the difference did not reach statistical significance, likely due to the limited sample size. We have included this observation in the results and acknowledged it as a potential source of bias in the discussion [Lines 260-263, 392-395].

Reviewer 1, major point #3:

I appreciate the effort to validate the FMF and NICE algorithms for predicting PET in a retrospective cohort of stillbirths versus matched controls. While MVM is present in all PET cases, MVM can also occur independently of PET, meaning that MVM can exist in non-hypertensive pregnancy disorders/placental dysfunction.

We appreciate this comment. We agree that MVM, while characteristic of PET, is not pathognomonic and can occur in other forms of placental dysfunction independent of maternal hypertension. We have clarified this point in the discussion and acknowledged that the presence of MVM in non-hypertensive stillbirths may limit the specificity of the FMF and NICE algorithms when applied retrospectively to such cohorts [Lines 354-358].

Reviewer 1, major point #4:

It would have been informative to examine placentas in livebirths of SGAs to explore whether MVM is present in these cases as well, particularly because approximately half of your stillbirths fetal weight is <10th centile. Although I appreciate it may not have been possible due to paucity of placentas, I believe authors should comment on literature findings of IUGR/SGAs placentas in regard to their findings.

We thank the reviewer for this insightful suggestion. We agree that comparison with placentas from liveborn SGA infants would have provided additional context particularly given that a substantial proportion of our stillbirths had fetal weights below the 10th centile. Unfortunately, placentas from preterm or SGA livebirths are only submitted for histopathological examination in our unit when specific clinical conditions are present such as chorioamnionitis or abruption which limited their availability for inclusion. We have now acknowledged this point in the Discussion and added a comment summarising the relevant literature indicating that MVM lesions are also common in SGA/IUGR pregnancies even in the absence of hypertension. This supports the concept that MVM reflects a shared pathway of placental dysfunction across different pregnancy outcomes. [Lines 400-407].

Reviewer 1, major point #5:

Twin and triplet pregnancies carry an increased risk of stillbirth per se, which may be further affected by chorionicity. Ideally, these cases should be analyzed separately; however, I understand that this may have been challenging. Please cite this as a potential source of bias and clarify whether "stillbirth" refers to the loss of one or both twins in the dataset.

We thank the reviewer for highlighting the potential influence of multiple pregnancies on stillbirth risk. We have now added a statement in the Discussion acknowledging that twin and triplet pregnancies may carry an increased risk of stillbirth, which could be further affected by chorionicity, and that analysing these cases separately was not feasible in our cohort [Lines 414-417]. We have also clarified in the Methods section that “stillbirth” refers to the loss of any fetus in a multiple pregnancy, and we have cited this as a potential source of bias [Lines 157-158].

Reviewer 1, minor point #1:

Please state clearly it is a retrospective case-control study, both in abstract and in manuscript.

We have clarified throughout the manuscript, including the title and abstract, that this study is a case-control study [Lines 1 and 26].

Reviewer 1, minor point #2:

The FMF algorithm outperformed NICE in predicting PET only when at least one continuous variable (e.g., MAP, UtA-PI, or PAPP-A) was included. In the absence of continuous variables, FMF and NICE are basically the same in terms of purely history-indicated PET screening. Please clarify both in abstract and manuscript.

We have clarified in both the abstract and the manuscript that the FMF algorithm outperformed NICE guidelines in predicting pre-eclampsia (PET) only when at least one continuous variable (e.g., MAP, UtA-PI, or PAPP-A) was included. In the absence of continuous variables, the FMF and NICE approaches are comparable for purely history-based PET screening [Lines 36-37; 362-364].

Reviewer 1, minor point #3:

line 16, 'placental disorders' should be changed to placental histopathological lesions.

We appreciate this helpful observation. We have updated line 16 to replace ‘placental disorders’ with ‘placental histopathological lesions’ as requested [Line 63].

Reviewer 2

We sincerely thank R 2 for the positive and thoughtful feedback. We appreciate your recognition of the study’s design, methodological strengths and clinical relevance.

Reviewer 2, point #1:

Inherent Limitations of Retrospective Design: The study is constrained by data completeness. As the authors note, the retrospective "reconstruction" of the FMF risk score was necessary due to missing first-trimester biophysical and biochemical markers in many cases, which may have impacted the accuracy of its calculated performance. Furthermore, clinical interventions during the study period (e.g., aspirin administration based on NICE criteria) could introduce a bias that is difficult to account for retrospectively

Thank you for this valuable comment. We fully acknowledge the inherent limitations of the retrospective design, including incomplete data and the need to reconstruct the FMF risk score. We have clarified these points in the revised manuscript and highlighted that any potential bias from contemporaneous clinical interventions such as aspirin use is discussed as a study limitation [Lines 424-427].

Reviewer 2, point #2:

Challenges in Control Group Selection: The authors rightly identify control selection as a major challenge. A perfect "low-risk" control group for stillbirths is difficult to obtain, as placentas from uncomplicated pregnancies are not routinely sent for pathology.

We thank R2 for highlighting this important point. We agree that control group selection represents a key limitation, as placentas from uncomplicated pregnancies are seldom available for pathological examination.

Reviewer 2, point #3:

Limited Sample Size: The total sample size of 83 stillbirths is relatively small for subgroup analyses, such as predicting PET (n=15) or specific MVM types. This can limit statistical power and leads to wide confidence intervals in some analyses, reducing the certainty of the findings.

Thank you for this observation. We agree that the relatively small sample size limits the power of subgroup analyses and contributes to wider confidence intervals. As noted in the manuscript, this limitation reflects the low incidence of stillbirth even in a large tertiary centre and highlights the need for larger multicentre studies to validate these findings.

We hope that these amendments effectively address the reviewers' concerns and suggestions.

Best regards,

Brenda on behalf of all co-authors

---

## [Decision Letter · Decision Letter 1]

25 Nov 2025

Placental lesions in stillbirths: a  case-control study using the Amsterdam criteria and predictive models at a UK tertiary unit

PONE-D-25-32249R1

Dear Dr. Narice,

We’re pleased to inform you that your manuscript has been judged scientifically suitable for publication and will be formally accepted for publication once it meets all outstanding technical requirements.

Kind regards,

Nishel Mohan Shah, PhD

Academic Editor

PLOS ONE

Additional Editor Comments (optional):

Reviewers' comments:

Reviewer's Responses to Questions

**Comments to the Author**

Reviewer #1: All comments have been addressed

Reviewer #2: All comments have been addressed

2. Is the manuscript technically sound, and do the data support the conclusions?

Reviewer #1: Yes

Reviewer #2: Yes

3. Has the statistical analysis been performed appropriately and rigorously?

Reviewer #1: Yes

Reviewer #2: Yes

4. Have the authors made all data underlying the findings in their manuscript fully available?

Reviewer #1: Yes

Reviewer #2: Yes

5. Is the manuscript presented in an intelligible fashion and written in standard English?

Reviewer #1: Yes

Reviewer #2: Yes

Reviewer #1: Thank you for your detailed response.

All the comments have been addressed. The revised manuscript has now expanded on potential sources of bias and has provided clarity on some specific issue.

Despite the retrospective nature and the limited size sample, the manuscript is scientifically sound and provides useful insight on histopathological lesions of stillbirth.

Reviewer #2: This is a well-designed and rigorously conducted retrospective case-control study. The study's core value lies in its dual focus: it not only identifies Maternal Vascular Malperfusion (MVM) as a dominant placental pathology in stillbirth using the standardized Amsterdam criteria, but also links this pathological finding to clinically relevant predictive tools (the FMF algorithm/Tommy's app). The findings strongly support a clinical shift from traditional NICE criteria to the more sensitive FMF algorithm for early-gestation risk identification, which could potentially lower stillbirth rates through interventions like aspirin the use of the standardized Amsterdam criteria for placental classification and the clinically relevant head-to-head comparison of the FMF and NICE predictive models.

The authors have provided excellent responses to editor and reviewer comments , thoroughly addressing limitations such as sample size, the retrospective design, and control group selection. While minor grammatical and formatting errors are present, the scientific core of the manuscript is solid and makes a valuable contribution to the field.

I recommend acceptance.

**Do you want your identity to be public for this peer review?** For information about this choice, including consent withdrawal, please see our Privacy Policy

Reviewer #1: No

Reviewer #2: No

---

## [Editor Report · Acceptance letter]

PONE-D-25-32249R1

PLOS One

Dear Dr. Narice,

I'm pleased to inform you that your manuscript has been deemed suitable for publication in PLOS One. Congratulations! Your manuscript is now being handed over to our production team.

Kind regards,

on behalf of

Dr. Nishel Mohan Shah

Academic Editor

PLOS One